# Laparoscopic Repeat Liver Resection—Selecting the Best Approach for Repeat Liver Resection

**DOI:** 10.3390/cancers15020421

**Published:** 2023-01-09

**Authors:** Zenichi Morise, Hidetoshi Katsuno, Kenji Kikuchi, Tomoyoshi Endo, Kazuhiro Matsuo, Yukio Asano, Akihiko Horiguchi

**Affiliations:** 1Department of Surgery, Fujita Health University School of Medicine Okazaki Medical Center, Okazaki 444-0827, Aichi, Japan; 2Department of Gastroenterological Surgery, Fujita Health University Bantane Hospital, Nagoya 454-8509, Aichi, Japan

**Keywords:** laparoscopic liver resection, repeat liver resection, robotic assisted liver resection, short-term outcome, liver function, simulation, navigation, anatomical liver resection

## Abstract

**Simple Summary:**

Intrahepatic cancer recurrence after liver resection is often treated with repeat resection, since it is considered the only curative treatment. However, it is associated with increased risk of complications. Laparoscopic liver resection for repeat resection is an emerging trend. Our multi-institutional propensity-score matching analyses of laparoscopic vs. open repeat liver resections for hepatocellular carcinoma showed feasibility and short-term advantages for selected patients undergoing laparoscopic repeat liver resection with comparable long-term outcomes. There are several disadvantages of laparoscopic repeat liver resection, including disorientation and the difficulty of repeated wide-range dissection of the Glissonian pedicles. Recently emerging small anatomical resection, indocyanine green fluorescence-guided surgery, and robot-assisted surgery are promising tools for the future development of laparoscopic repeat liver resection. The present review discusses how laparoscopic repeat liver resection, as a powerful unique local therapy causing less damage to the residual liver and surrounding structures, could contribute to the outcomes of repeat treatments for cancers and its future perspectives.

**Abstract:**

Recurrence of liver cancers after liver resection (LR), such as recurrences of hepatocellular carcinoma and colorectal liver metastases, is often treated with repeat LR (RLR) as the only curative treatment. However, RLR is associated with an increased risk of complications. The indications for the currently emerging laparoscopic LR and its advantages and disadvantages for repeat treatment are still under discussion. Our multi-institutional propensity-score matched analyses of laparoscopic vs. open RLRs for hepatocellular carcinoma showed the feasibility of laparoscopic RLR with comparable short- and long-term outcomes. Small blood loss and low morbidity was observed in selected patients treated using laparoscopic RLR in which total adhesiolysis can be dodged, with speculations that laparoscopic minor repeated LR can minimize functional deterioration of the liver. However, there are several disadvantages, such as easily occurring disorientation and difficulty in repeated wide-range dissection of Glissonian pedicles. Recently emerging small anatomical resection, indocyanine green fluorescence-guided surgery, and robot-assisted surgery are promising tools for the further development of laparoscopic RLR. This review discusses how laparoscopic RLR, as a powerful unique local therapy causing less damage to the residual liver and surrounding structures, could contribute to the outcomes of repeated treatments for cancers and its future perspectives.

## 1. Introduction

Liver resection (LR) is performed mainly in case of patients with hepatocellular carcinoma (HCC), liver metastasis of colorectal carcinoma (CRCLM), or biliary tract carcinomas as cancer treatment [1,2,3]. Patients with HCC and CRCLM often undergo repeat treatments after the first LR for intrahepatic recurrent lesions in the absence of or well-controlled extrahepatic metastases [1,4]. These surgeries are performed as curative intent therapy or pursuing long-term survival without other effective alternatives, except for only a small proportion of patients who can undergo potentially curative treatments of ablation therapy for small HCC and liver transplantation for non-advanced HCC with severe cirrhosis [1]. However, in repeat LR (RLR), adhesion disturbs the mobilization of the liver and the dissection of vasculatures, and scars/adhesions cause deformity, making the identification of intrahepatic structures and tumors difficult. This can lead to intra- or post-operative complications [5,6,7].

Laparoscopic LR (LLR) has been an emerging procedure for the last three decades [8]. Its indications have expanded rapidly, and reports on laparoscopic repeat LR (LRLR) are increasing. We conducted retrospective international multi-institutional studies of LRLR [9,10,11]. The studies were conducted in association with the 2nd Congress of International Laparoscopic Liver Society (ILLS-Tokyo, 2019 [12]) and 1582 HCC cases with RLR were registered from 42 prominent international liver surgery centers (Asian 25, European 14, American 2, Australian 1). In the present review, we summarize the results of the studies and discuss the advantages and disadvantages of the procedure using our own data, as well as future perspectives, including the possible merits of robot-assisted procedures.

## 2. Summary of ILLS-Tokyo International Multi-Institutional Studies of LRLR for HCC

A series of three studies [9,10,11] was published by the ILLS-Tokyo LRLR collaboration. 

The main study [9] was the first and largest (1582 cases) international propensity score matching study that compared LRLR and open RLR for HCC patients. The results showed that LRLR was feasible and comparable to open RLR in terms of both short- and long-term outcomes. Overall conversion rate of LRLR was 3.8% under the patient selection at each institution. LRLR was preferably adopted in patients with poor general/liver condition but was more favorable in tumor and surgical procedure-related conditions. There were notable differences between the centers in terms of LRLR indication. The number of LRLR cases in each center ranged from 0 to 67 (median, 10), and the percent of LRLR in all RLR cases ranged from 0 to 100 (median, 57.1%) in each center, and no correlation was observed between the number and percentage (*p* = 0.349). Each center seemed to apply the laparoscopic procedure for recurrent HCC patients planned to undergo RLR depending on their own experience based on the difference in prevalence of HCC in each region. This indicates that this procedure is still in its developmental stage, but not a common surgical procedure, even among high-volume LR centers worldwide at the time. Propensity score-based matching was performed using the indicators related to patients’ background condition, liver function, tumor condition, and surgical procedure, and 238:238 matched patients were obtained for the LRLR and open RLR groups with well-balanced indicators. Short-term outcomes were comparable, with a slightly longer operation time (LRLR vs. open RLR, 272 ± 187 min vs. 232 ± 129 min, *p* = 0.007) and much smaller blood loss (268 ± 730 vs. 497 ± 784 mL, *p* = 0.001) in the LRLR group. Although this study showed that LRLR was comparable to the open procedure for short-term outcomes, decreased morbidity has been considered as one of the advantages of LLR for HCC patients with chronic liver diseases [13,14,15,16]. The hypothesis that this analysis included complicated cases of LRLR that were performed during the development stage of the procedure and, therefore, failed to show the advantage, was examined in the following study [10]. In the second study, cases with tumors >1 cm from the major vessels, defined as less complicated cases, were analyzed in the same way as in the first study. This study with 115 each matched patients showed less blood loss (283 ± 823 mL vs. 604 ± 665 mL, *p* = 0.001) and less morbidity (≥Clavien-Dindo grade (CD) II, 8.7 % vs. 18.3%, *p* = 0.034; ≥CD III, 4.3% vs. 12.2%, *p* = 0.031) in LRLR group. Operation times were similar (260.6 ± 158.3 vs. 270.0 ± 129.6 min, *p* = 0.622). The length of post-operative hospital stay was shorter in the LRLR group (10.2 ± 11.3 vs. 13.2 ± 12.1 days, *p* = 0.058), but not significant due to the large difference between centers, which may be due to differences in insurance systems and hospitalization practices in their regions and countries. The hypothesis mentioned above is supported by the results of this second study, and it is believed that LRLR can become advantageous even for treating complicated cases when it is established as the common surgical procedure.

In the first study of matching analysis [9], there were no significant differences in overall or disease-free survival time between the LRLR and open RLR groups. However, the curves of overall survival were clearly separated with better tendency in LRLR (medians overall survival of LRLR vs. open RLR:12.55 vs. 8.94 years, *p* = 0.0855), although the curves of disease-free survival were identical and overlapped (*p* = 0.517). The overall survival after LR for HCC patients with chronic liver diseases was determined not only by the oncological effect on the existing HCC but also by the treatments for future multicentric new HCC and residual liver function. Adequate residual liver function is needed for repeated treatment and long-term survival without liver insufficiency. It is speculated that overall survival after LRLR was better, since there was less damage to liver function in LRLR [13,14], making the repeat treatments more accessible and reducing the number of deaths due to liver insufficiency. However, there is currently no effective tool for assessing chronic liver functional deterioration after LR. Therefore, data on liver functional indicators from 657 patients who underwent segment or less resection were evaluated to identify candidates that could detect permanently fixed chronic liver functional deterioration 3 months after RLR in the third study. Patients with a segment or less resection were selected to exclude the impact of the reduction in functional liver volume after resection. In the comparison between the values before and 3 months after RLR, plasma albumin level (4.04 ± 0.45 vs. 3.97 ± 0.53 g/dL, *p* = 0.006), total bilirubin level (0.76 ± 0.33 vs. 0.81 ± 0.40 mg/dL, *p* = 0.01), and ALBI score (−2.73 ± 0.40 vs. −2.65 ±−0.48, *p* = 0.001) showed significant changes, indicating liver functional deterioration after surgery. These indicators could be used in future investigations of chronically fixed liver functional deterioration after LR. 

Two-hundred and sixty-eight patients who underwent segment or less open RLR after open LR and 224 patients who underwent segment or less LRLR after LLR were compared regarding their backgrounds and the changes in albumin, total bilirubin and ALBI scores before and 3 months after LR in the third study as well. Before RLR, a significantly higher BMI (*p* = 0.002), poorer performance status (*p* = 0.043), higher incidence of liver fibrosis (*p* = 0.006), and higher Child–Pugh score (Child–Pugh score in LRLR vs. open RLR: 160:53:7:4 (5:6:7:>8) vs. 239:25:4:0, *p* < 0.001) in the LRLR group, with no significant differences in tumor- and surgery-related factors. Although there were significant differences in plasma levels of albumin (LRLR vs. open RLR before RLR, *p* < 0.001; LRLR vs. open RLR after RLR, *p* = 0.003), blood platelet count (*p* = 0.031; *p* = 0.025) and ALBI score (*p* < 0.001; *p* = 0.003) between the groups at both time points before and after RLR, all the changes of values between the time points before and after RLR in albumin (changes in LRLR vs. open RLR:0.054 ± 0.42 vs. 0.068 ± 0.40 g/dL, *p* = 0.710), bilirubin (−0.049 ± 0.33 vs. −0.036 ± 0.34 mg/dL, *p* = 0.653) and ALBI score (−0.063 ± 0.38 vs. −0.064 ± 0.35, *p* = 0.969) were similar without significant differences between the groups. Although the patients in the LRLR group had poor liver function (plus general condition) before RLR, they underwent RLR with comparable deterioration in liver function to that of open patients with better liver function.

## 3. Advantages and Disadvantages of LRLR and Our Experience

Anatomical/structural modifications and adhesions after surgery increase the difficulty of performing the surgery. In RLR, adhesions disturb mobilization of the liver and the dissection of vasculatures, and scars/adhesions cause deformity, which makes the identification of intra-hepatic structures and tumors difficult. They can lead to intra- and post-operative complications. It has been advocated that the laparoscopic approach makes the following redo surgeries easier due to the reduction of adhesion-formation [17]. Moreover, there is another specific advantage of LRLR. The “caudal approach” of LLR (Figure 1), which we presented as novel concept in 2013 [18], was defined as a main conceptual change in LLR in the Second International Consensus Conference of LLR [19]. LR is a procedure of resecting the liver protected inside the subphrenic space (“rib cage”), and conventional open procedures are performed with a large subcostal incision and mobilization of the liver. In open LR, the “cage” is destructively opened, and the liver is taken out from the “cage” with mobilization and compression. In LLR, contrarily, the procedure is performed with minimum damage to the “cage” and residual liver, by direct intrusion of laparoscope and instruments into the cage and the operative field from caudal direction. This laparoscopic-specific approach has advantages for patients with chronic liver diseases, reduction of post-operative ascites, and liver failure due to less damage to the surrounding structures, including collateral vessels and residual liver [13]. This LLR-specific “caudal approach” can also work favorably for RLR. When adequate access to the lesion and sufficient working space for surgery are obtained, LRLR can be performed even in smaller spaces such as those between adhesions. Although total adhesiolysis of the entire upper abdomen around the liver is performed during the first part of open RLR with laparotomy, it is not always necessary in LRLR, particularly when the target lesion is within the surface area of the liver (Figure 1). In our own experience, there were no significant increases in operation time and intraoperative blood loss in LRLR compared with the first LLR (Table 1). This is a much different situation from open RLR, since total adhesiolysis for the open procedure takes more time and leads to more blood loss compared to that in primary surgery. When the plan to obtain good access to the target area and adequate working space between adhesions can be established under precise preoperative simulation, LRLR could have this advantage over open RLR using intraoperative navigation (Figure 2). LRLR with a specific caudal approach is a unique and strong local treatment that can be applied repeatedly. It has conceptual benefits, such as in patients with HCC/chronic liver disease (CLD) who often develop repeated metachronous lesions from an oncogenic background of a chronically injured liver. 

However, LLR has certain disadvantages. Disorientation can easily occur due to the lack of fine haptics, lack of an overview of the entire operative field, and difficulty in performing precise intraoperative ultrasonography. Simulation and navigation from pre- and intra-operative imaging studies have also been used to overcome these disadvantages. The Tokyo 2020 terminology for liver anatomy and resections was recently issued for segment and/or smaller anatomical resections [20]. Although these small anatomical resections and their combinations are under discussion for their oncological advantages, there are advocated benefits for small anatomical (“cone-unit”) resections. It can secure the tumor location with a sufficient resection margin in the resected area under precise preoperative simulation and navigation (Figure 2). It could also lead to less postoperative bile leakage and less residual ischemia/congestive parenchyma, which may lead to recurrence [21]. 

We experienced 48 LRLR cases (LRLR group), including 9 third-, 3 fourth-, and 2 fifth-times repeated LR. There were no cases of three segments or more LR in the group. During the same period, 129 patients (LPLR group) underwent laparoscopic primary LR of less than 3 segments. Table 1 presents the details of the groups. Among the background factors, LRLR had more HCC cases (*p* = 0.008), smaller tumors (*p* = 0.021), smaller extent of resection (*p* = 0.037), and poorer liver function (indicated by ICGR15 value, *p* = 0.035; prothrombin time, *p* = 0.034; platelet count, *p* < 0.001), although sex, age, and tumor location (anterolateral or posterosuperior segments) were similar. In the short-term outcomes, there were no significant differences in operation time (LPLR vs. LRLR, 330.16 ± 140.23 vs. 309.79 ± 147.02 min, *p* = 0.409), intraoperative blood loss (226.72 ± 356.79 vs. 275.15 ± 553.20 mL, *p* = 0.495), blood transfusion rate (109:20 (no: yes) vs. 38:10, *p* = 0.401), conversion rate (128:1 vs. 46:2, *p* = 0.179), postoperative morbidity of Clavien–Dindo grade 3 or above (121:8 vs. 43:5, *p* = 0.343) and hospital stay (17.26 ± 13.51 vs. 18.47 ± 18.65 days, *p* = 0.690). To match the background factors, patients with HCC and a segment or less resection, which formed a majority of the LRLR group, were extracted. Forty-two case group of primary LR (LPLR-HCC group) and 34-case group of repeat LR (LRLR-HCC group) were identified (Table 2). There were no significant differences in background factors between the groups, except for a lower platelet count in the LRLR-HCC group (*p* = 0.031). Thereafter, there were still no significant differences in all the indicators of short-term outcomes between the groups (operation time, 287.40 ± 117.91 vs. 272.68 ± 128.034 min, *p* = 0.608; intraoperative blood loss 141.86 ± 217.69 vs. 284.09 ± 641.81 mL, *p* = 0.183; blood transfusion (no: yes), 40:2 vs. 28:6, *p* = 0.129; conversion to laparotomy, 42:0 vs. 33:1, *p* = 0.447; morbidity, 41:1 vs. 30:4, *p* = 0.167; postoperative hospital stay, 15.36 ± 8.85 vs. 19.41 ± 21.41 days, *p* = 0.320). Open repeat LR usually requires more operative time and causes more blood loss due to adhesions compared to primary surgery. The outcomes of LRLR from our experience are much different from those of open surgery. Less than segment LRLR for HCC patients was performed with a similar operation time and blood loss to primary LLR with similar backgrounds. We believe this was caused by the conceptual advantages of the “caudal approach” mentioned above. However, intra- and post-operative complications were experienced during repeated and wide-range dissection of the hilar area of the Glissonian pedicle. One of the fifth-time repeated LLR cases converted to laparotomy because of bleeding from the dorsal area of the dissected hilar (to segments 8 and 1) Glissonian pedicle. This patient had undergone a third-time repeated dissection of the hilar area at the time. The working space for surgery was relatively limited by the stiff fibrosis around the area. In these conditions, wider dissection is needed and, therefore, the advantage of LRLR might be limited. In addition, in these conditions, robot-assisted LRLR might have an advantage, even without CUSA for parenchymal transection [22,23].

## 4. Indications of LRLR

One of the main obstacles to LRLR indication is adhesion, which usually occurs around the liver-resected area, port sites, and other dissected areas. Adhesion in the liver-resected area is the thickest and the most difficult to resolve. Port placement and the plan for the access route are arranged preoperatively under the prediction of adhesion with the information of previous surgeries and imaging studies (Figure 2). In this manner, most LRLR procedures can be safely performed. Although the indication of LRLR is varied even between high-volume centers as mentioned before, LRLR is adopted under the same indication as primary LLR in our institution. 

However, as mentioned above, repeated wide-range dissection of the hilar area of the Glissonian pedicle is difficult. Although one of the advantages of LRLR is that it does not require total adhesiolysis, it should be extended in order to acquire enough and safe working space between the stiff scars and adhesion in those cases. In addition, size of tumors should be an additional limitation for indication when extended working space is needed for large tumor. However, there were no large tumors experienced in our institution under the close follow up after primary surgery (Table 1. 60 mm and 24 mm were the maximum and mean sizes of the tumors.). The indication for LRLR in these cases should be judged by balancing its advantages and disadvantages. In addition, LRLR with vessel resection and reconstruction is currently out of our indication. 

Patients with HCC or CRCLM are candidates for LRLR. However, these two diseases have different backgrounds. Patients with HCC mostly have CLD, which causes functional deterioration and multicentric oncogenicity in the injured liver [1,24]. LRLR for HCC is performed in fibrotic and atrophic livers with deteriorated function. It is sometimes difficult to identify small tumors among degenerative nodules in cirrhotic livers, especially in LRLR with liver deformation. Since patients develop collateral vessels even in adhesions, difficulty in bleeding control is often experienced accompanied with cirrhotic coagulopathy. Destruction of collateral vessels can lead to massive ascites and liver failure. In contrast, CRCLM occurs in the liver without CLD, but is affected by chemotherapy in most cases [24]. Congestive or steatotic liver after chemotherapy is easy-to-bleed and fragile during surgery. Small lesions that shrink after chemotherapy are difficult to identify, especially those with adhesions and scars around the liver. The indications for LRLR for these diseases are the same as those for primary LLR; however, the disease-specific conditions mentioned above could be additional obstacles in the procedure and may change the indications.

## 5. Future Perspectives

LLR is advancing into new techniques, such as hepatic vein-guided small anatomical resection [25], indocyanine green (ICG) fluorescence-guided anatomical resection and tumor identification [26], LLR with a Glissonian approach to peripheral small branches (“cone unit” resection) [27], and robot-assisted LLR [22,23]. These techniques are predicted to work in the difficult LRLR cases mentioned above. Novel anti-adherent agents, which ease adhesion and make dissection and adhesiolysis less difficult in LRLR, are being developed [28]. These are all promising tools for LRLR.

## 6. Conclusions

LRLR with conceptual benefits from its specific "caudal approach" is a unique and powerful local therapy that can be applied repeatedly for intrahepatic recurrences of cancers. Although disorientation can occur and adhesion and scars can be obstacles, they can be conquered by well-planed procedure with preoperative simulation and navigation.

## Figures and Tables

**Figure 1 cancers-15-00421-f001:**
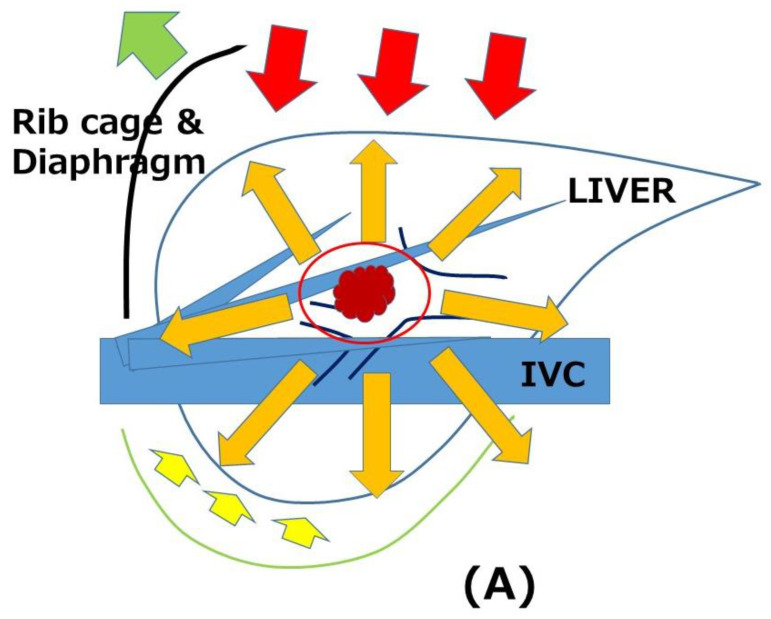
Open (**A**) and laparoscopic “caudal approach” (**B**) repeat liver resections. The directions of view and manipulation in each approach are indicated with red arrows. (**A**) In the open approach, the subphrenic rib cage was opened with a large subcostal incision and the liver was mobilized (lifted) from the retroperitoneum. (**B**) In laparoscopic approach, the instruments were introduced into the cage from the caudal direction and the surgery was performed with minimal damage to the associated structures. Orange arrows indicate the dissection of adhesion. (**A**) Total adhesiolysis was performed in open procedure. (**B**) Direct approach (also indicated with orange arrows) to the tumor in laparoscopic “caudal approach” can facilitate surface repeat liver resection with minimal adhesiolysis.

**Figure 2 cancers-15-00421-f002:**
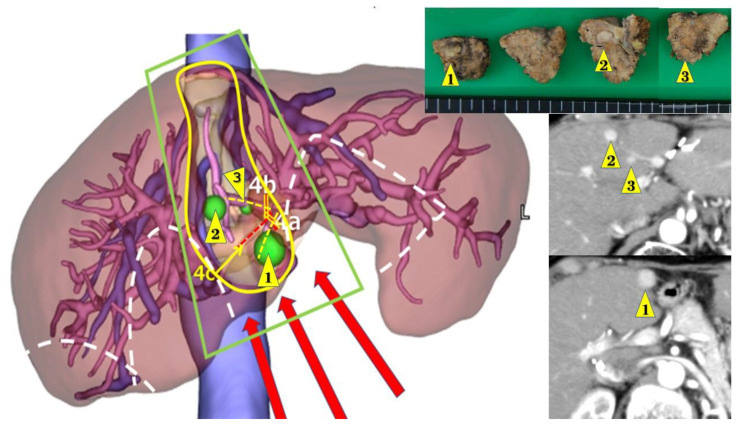
Simulation for fourth laparoscopic liver resection of three tumors in segment 4. Three tumors (1,2, and 3) all developed in segment 4 after segment 3 segmentectomy (1st laparoscopic liver resection (LLR)), partial resection of segments 5–6 (2nd LLR) and partial resection of segments 1–7 (3rd LLR). Tumor 1 was in caudal area near the border of segments 4–3. Tumor 2 was in cranial surface area. Tumor 3 was in cranial dorsal area. Left: preoperative CT simulation from CT reconstruction. The area encircled with yellow line, which is the portal area of Glissonian pedicles 4a and 4b (yellow ruptured lines) and includes three tumors to be resected. Glissonian pedicle 4c (red ruptured line) is planned to be exposed and preserved on the bottom surface of transection. The areas encircled by white ruptured lines show the adhesion from previous surgeries. With the planned approach route shown with red arrows, the area encircled with green line is the planned surgical working space secured with minimum adhesiolysis. Right upper: resected specimen. Right lower: preoperative findings of computed tomography with contrast.

**Table 1 cancers-15-00421-t001:** Comparison between laparoscopic primary and repeat liver resection in terms of background factors and postoperative short-term outcomes.

	*LPLR Group*, n = 129	*LRLR Group*, n = 48	*p* Value
**Background Factors**			
**Age (years old)**	67.47 ± 10.84	70.19 ± 7.60	0.102
**Sex (Male: Female)**	77:52	32:16	0.396
**Diseases for LR (HCC: Mets: other)**	60:46:23	34:12:2	0.008 *
**ICG R15 (%)**	14.41 ± 12.29	18.66 ± 10.51	0.035 *
**Total Bilirubin (mg/dL)**	0.71 ± 0.34	0.78 ± 0.31	0.171
**Prothrombin Time (%)**	98.63 ± 15.87	93.10 ± 14.94	0.034 *
**Albumin (g/dL)**	3.93 ± 0.47	3.78 ± 0.48	0.076
**Platelet (×10^4^/microl)**	19.72 ± 8.77	14.24 ± 6.93	<0.001 *
**Number of tumors**	1.36 ± 0.77	1.42 ± 0.79	0.669
**Size of tumor (mm)**	32.64 ± 25.47	23.75 ± 11.03	0.021 *
**Tumor location (AL:PS)**	64:65	24:24	0.818
**Extent of resection** **(Pt:LLS:Seg:Sec)**	82:9:12:26	36:2:8:2	0.037 *
**Short-term Outcomes**			
**Operation time (minutes)**	330.16 ± 140.23	309.79 ± 147.02	0.409
**Intraoperative Blood loss (mL)**	226.72 ± 356.79	275.15 ± 553.20	0.495
**Blood transfusion (no: yes)**	109:20	38:10	0.401
**Conversion to laparotomy (no: yes)**	128:1	46:2	0.179
**Morbidity (no: yes)**	121:8	43:5	0.343
**Postoperative hospital stay (day)**	17.26 ± 13.51	18.47 ± 18.65	0.690

* Statistically significant. LPLR, laparoscopic primary liver resection; LRLR, laparoscopic repeat liver resection LR, liver resection; HCC, hepatocellular carcinoma; Mets, liver metastasis; ICGR15, indocyanine green retention at 15 min; AL, anterolateral segments = segments 2,3,4b,5,6; PS, posterosuperior segments = 1,4a,7,8; Pt, partial resection; LLS, left lateral sectionectomy; Seg, segmentectomy; Sec, sectionectomy; Morbidity, Clavien–Dindo grade 3 or above.

**Table 2 cancers-15-00421-t002:** Comparison between laparoscopic primary and repeat liver resection for the HCC patients who have undergone segmentectomy or less resection in background factors and postoperative short-term outcomes.

	*LPLR-HCC Group*, n = 42	*LRLR-HCC Group*, n = 34	*p* Value
**Background Factors**			
**Age (years old)**	70.29 ± 8.67	71.38 ± 7.68	0.561
**Sex (Male: Female)**	14:28	13:21	0.657
**ICG R15 (%)**	20.27 ± 15.31	20.63 ± 10.71	0.905
**Total Bilirubin (mg/dL)**	0.94 ± 0.42	0.82 ± 0.33	0.167
**Prothrombin Time (%)**	90.17 ± 14.06	89.56 ± 13.86	0.853
**Albumin (g/dL)**	3.81 ± 0.52	3.71 ± 0.53	0.435
**Platelet (** **×10^4^/microl)**	14.62 ± 7.84	11.32 ± 4.38	0.031 *
**Number of tumors**	1.19 ± 0.51	1.38 ± 0.70	0.169
**Size of tumor (mm)**	27.12 ± 15.73	21.82 ± 8.60	0.067
**Tumor location (AL:PS)**	25:17	20:15	0.951
**Short-term Outcomes**			
**Operation time (minutes)**	287.40 ± 117.91	272.68 ± 128.034	0.608
**Intraoperative Blood loss (mL)**	141.86 ± 217.69	284.09 ± 641.81	0.183
**Blood transfusion (no: yes)**	40:2	28:6	0.129
**Conversion to laparotomy (no: yes)**	42:0	33:1	0.447
**Morbidity (no: yes)**	41:1	30:4	0.167
**Postoperative hospital stay (day)**	15.36 ± 8.85	19.41 ± 21.41	0.320

* Statistically significant; LPLR, laparoscopic primary liver resection; LRLR, laparoscopic repeat liver resection; LR, liver resection; HCC, hepatocellular carcinoma; Mets, liver metastasis; ICGR15, indocyanine green retention at 15 min; AL, anterolateral segments = segments 2,3,4b,5,6; PS, posterosuperior segments = 1,4a,7,8; Pt, partial resection; LLS, left lateral sectionectomy; Seg, segmentectomy; Sec, sectionectomy; Morbidity, Clavien–Dindo grade 3 or above.

## Data Availability

The data presented in this study are available upon request from the corresponding author.

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
