# Peer review of "Laparoscopic Repeat Liver Resection—Selecting the Best Approach for Repeat Liver Resection"

_cancers, 2023, doi:10.3390/cancers15020421_

Round 1

Reviewer 1 Report

Thank you and congratulations for your concise review for lap repeat resection. 

Although there are many biases, recent reports show that LRLR is feasible and promising. 

As described in the manuscript, there were notable differences between centers in LRLR indication. It will better that you will summarize and give some guideline or recommendation for indication.

The main obstacle for LRLR will be adhesion. You can briefly add more information for prediction of adhesion preoperatively and how to quantify degree of adhesion.

 There are definite differences in LRLR between HCC and CRLM. You can briefly add differences.

Finally, what is your contraindication for LRLR? When open repeat resection is recommended?

Author Response

Dear Editors and Reviewers,

Thank the reviewers for their kind and instructive comments.

As mentioned below, I put the new chapter 4 INDICATIONS OF LRLR for the response to the reviewers’ comments. Also, external English-editing was performed as shown as certificate document attached.

Thank you again for your efforts handling our manuscript.

Best regards,

Zenichi Morise M.D., Ph.D., FACS

Professor and Chairman, Department of Surgery Fujita Health University School of Medicine

Founding Past Director, Fujita Health University Okazaki Medical Center

Deputy Chief Editor, Fujita Medical Journal

1 Gotanda Harisakicho,

Okazaki, AICHI 444-0827 JAPAN

Phone:+81-564-64-8800

FAX:+81-564-64-8135

Response to Reviewer 1

Thank you and congratulations for your concise review for lap repeat resection. Although there are many biases, recent reports show that LRLR is feasible and promising. 

Thank you for your kind and instructive comments.

As described in the manuscript, there were notable differences between centers in LRLR indication. It will better that you will summarize and give some guideline or recommendation for indication.

Response: We put newly added chapter 4 for LRLR indication copied below.

One of the main obstacles to LRLR indication is adhesion, which usually occurs around the liver-resected area, port sites, and other dissected areas. Adhesion in the liver-resected area is the thickest and the most difficult to resolve. Port placement and the plan for the access route are arranged preoperatively under the prediction of adhe-sion with the information of previous surgeries and imaging studies (Figure 2). In this manner, most LRLR procedures can be safely performed. Although the indication of LRLR is varied even between high-volume centers as mentioned before, LRLR is adopted under the same indication as primary LLR in our institution.

However, as mentioned above, repeated wide-range dissection of the hilar area of the Glissonian pedicle is difficult. Although one of the advantages of LRLR is that it does not require total adhesiolysis, it should be extended in order to acquire enough and safe working space between the siff scars and adhesion in those cases. Also, size of tumors should be an additional limitation for indication when extended working space is needed for large tumor. However, there were no large tumors experienced in our in-stitution under the close follow up after primary surgery (Table 1. 60 mm and 24 mm were the maximum and mean sizes of the tumors.). The indication for LRLR in these cases should be judged by balancing its advantages and disadvantages. In addition, LRLR with vessel resection and reconstruction is currently out of our indication.

Patients with HCC or CRCLM are candidates for LRLR. However, these two dis-eases have different backgrounds. Patients with HCC mostly have CLD, which causes functional deterioration and multicentric oncogenicity in the injured liver 24,25. LRLR for HCC is performed in fibrotic and atrophic livers with deteriorated function. It is some-times difficult to identify small tumors among degenerative nodules in cirrhotic livers, especially in LRLR with liver deformation. Since patients develop collateral vessels even in adhesions, difficulty in bleeding control is often experienced accompanied with cir-rhotic coagulopathy. Destruction of collateral vessels can lead to massive ascites and liver failure. In contrast, CRCLM occurs in the liver without CLD, but is affected by chemotherapy in most cases 25. Congestive or steatotic liver after chemotherapy is easy-to-bleed and fragile during surgery. Small lesions that shrink after chemotherapy are difficult to identify, especially those with adhesions and scars around the liver. The indications for LRLR for these diseases are the same as those for primary LLR; however, the disease-specific conditions mentioned above could be additional obstacles in the procedure and may change the indications.

The main obstacle for LRLR will be adhesion. You can briefly add more information for prediction of adhesion preoperatively and how to quantify degree of adhesion.

Response: We put sentences below in newly added chapter 4 for LRLR indication.

One of the main obstacles to LRLR indication is adhesion, which usually occurs around the liver-resected area, port sites, and other dissected areas. Adhesion in the liver-resected area is the thickest and the most difficult to resolve. Port placement and the plan for the access route are arranged preoperatively under the prediction of adhesion with the information of previous surgeries and imaging studies (Figure 2).

 There are definite differences in LRLR between HCC and CRLM. You can briefly add differences.

Response: We put sentences below in newly added chapter 4 for LRLR indication.

Patients with HCC mostly have CLD, which causes functional deterioration and multicentric oncogenicity in the injured liver 24,25. LRLR for HCC is performed in fibrotic and atrophic livers with deteriorated function. It is sometimes difficult to identify small tumors among degenerative nodules in cirrhotic livers, especially in LRLR with liver deformation. Since patients develop collateral vessels even in adhesions, difficulty in bleeding control is often experienced accompanied with cirrhotic coagulopathy. Destruction of collateral vessels can lead to massive ascites and liver failure.

In contrast, CRCLM occurs in the liver without CLD, but is affected by chemotherapy in most cases 25. Congestive or steatotic liver after chemotherapy is easy-to-bleed and fragile during surgery. Small lesions that shrink after chemotherapy are difficult to identify, especially those with adhesions and scars around the liver.

Finally, what is your contraindication for LRLR? When open repeat resection is recommended?

Response: We put sentences below in newly added chapter 4 for LRLR indication.

LRLR is adopted under the same indication as primary LLR in our institution.

However, as mentioned above, repeated wide-range dissection of the hilar area of the Glissonian pedicle is difficult. … Also, size of tumors should be an additional limitation for indication when extended working space is needed for large tumor. … The indication for LRLR in these cases should be judged by balancing its advantages and disadvantages.

In addition, LRLR with vessel resection and reconstruction is currently out of our indication.

The indications for LRLR for these diseases are the same as those for primary LLR; however, the disease-specific conditions mentioned above could be additional obstacles in the procedure and may change the indications.

Reviewer 2 Report

This is an interesting manuscript with comprehensive data on repeat laparoscopic liver resection for HCC. Extensive editing of English language should be performed. After improving the language, I think that this review can play a role in summarizing the existing evidence on the question of repeat liver resection for HCC.

Author Response

Dear Editors and Reviewers,

Thank the reviewers for their kind and instructive comments.

As mentioned below, I put the new chapter 4 INDICATIONS OF LRLR for the response to the reviewers’ comments. Also, external English-editing was performed as shown as certificate document attached.

Thank you again for your efforts handling our manuscript.

Best regards,

Zenichi Morise M.D., Ph.D., FACS

Professor and Chairman, Department of Surgery Fujita Health University School of Medicine

Founding Past Director, Fujita Health University Okazaki Medical Center

Deputy Chief Editor, Fujita Medical Journal

1 Gotanda Harisakicho,

Okazaki, AICHI 444-0827 JAPAN

Phone:+81-564-64-8800

FAX:+81-564-64-8135

Response to Reviewer 2

This is an interesting manuscript with comprehensive data on repeat laparoscopic liver resection for HCC. Extensive editing of English language should be performed. After improving the language, I think that this review can play a role in summarizing the existing evidence on the question of repeat liver resection for HCC.

Thank you for your kind and instructive comments.

External English-editing was performed as shown as certificate document attached.

Round 2

Reviewer 1 Report

Thank you for your revision.